# The Importance of RSV Epidemiological Surveillance: A Multicenter Observational Study of RSV Infection during the COVID-19 Pandemic

**DOI:** 10.3390/v15020280

**Published:** 2023-01-19

**Authors:** Giulia Pruccoli, Emanuele Castagno, Irene Raffaldi, Marco Denina, Elisa Barisone, Luca Baroero, Fabio Timeus, Ivana Rabbone, Alice Monzani, Gian Maria Terragni, Cristina Lovera, Adalberto Brach del Prever, Paolo Manzoni, Michelangelo Barbaglia, Luca Roasio, Simona De Franco, Carmelina Calitri, Maddalena Lupica, Enrico Felici, Cinzia Marciano, Savino Santovito, Gaia Militerno, Enrica Abrigo, Antonio Curtoni, Paola Quarello, Claudia Bondone, Silvia Garazzino

**Affiliations:** 1Infectious Diseases Unit, Department of Pediatrics, University of Turin, Regina Margherita Children’s Hospital, A.O.U. Città Della Salute e della Scienza di Torino, 10126 Turin, Italy; 2Department of Pediatric Emergency, Regina Margherita Children’s Hospital, A.O.U. Città Della Salute e della Scienza di Torino, 10126 Turin, Italy; 3Department of Public Health and Pediatrics, University of Turin, 10126 Turin, Italy; 4Department of Pediatrics, Martini Hospital, 10141 Turin, Italy; 5Pediatrics Department, Chivasso Hospital, 10034 Chivasso (TO), Italy; 6Division of Pediatrics, Department of Health Sciences, University of Piemonte Orientale, 28100 Novara, Italy; 7Department of Pediatrics, Chieri Hospital, 10023 Chieri (TO), Italy; 8Department of Pediatrics, A.O. S.Croce e Carle, 12100 Cuneo, Italy; 9Department of Pediatrics, Presidio Ospedaliero di Ciriè, ASL TO4, 10073 Ciriè (TO), Italy; 10Department of Pediatrics, Ospedale Degli Infermi di Ponderano, University of Turin, 13900 Biella, Italy; 11Department of Pediatrics, Ospedale Castelli, 28923 Verbania, Italy; 12Department of Pediatrics, Edoardo Agnelli Hospital, 10064 Pinerolo (TO), Italy; 13Department of Pediatrics, Ospedale di Borgomanero, 28021 Borgomanero (NO), Italy; 14Department of Pediatrics, Ospedale di Rivoli, 10098 Rivoli (TO), Italy; 15Pediatric and Pediatric Emergency Unit, Children’s Hospital, AO SS Antonio e Biagio e C. Arrigo, 15121 Alessandria, Italy; 16Department of Pediatrics, Ospedale Maria Vittoria, ASL Città di Torino, 10143 Turin, Italy; 17Department of Pediatrics, Ospedale Cardinal Massaia, 14100 Asti, Italy; 18Microbiology and Virology Unit, Molinette Hospital, A.O.U. Città della Salute e della Scienza di Torino, 10126 Turin, Italy; 19Department of Pediatric Onco-Hematology, University of Turin, Regina Margherita Children’s Hospital, A.O.U. Città Della Salute e della Scienza di Torino, 10126 Turin, Italy

**Keywords:** respiratory syncytial virus (RSV), epidemiology, coinfections, COVID-19 pandemic, seasonal peak, epidemiological surveillance

## Abstract

The restrictive measures adopted worldwide against SARS-CoV-2 produced a drastic reduction in respiratory pathogens, including RSV, but a dramatic rebound was thereafter reported. In this multicenter retrospective observational study in 15 Pediatric Emergency Departments, all children <3 years old with RSV infection admitted between 1 September and 31 December 2021 were included and compared to those admitted in the same period of 2020 and 2019. The primary aim was to evaluate RSV epidemiology during and after the COVID-19 pandemic peak. The secondary aims were to evaluate the clinical features of children with RSV infection. Overall, 1015 children were enrolled: 100 in 2019, 3 in 2020 and 912 in 2021. In 2019, the peak was recorded in December, and in 2021, it was recorded in November. Comparing 2019 to 2021, in 2021 the median age was significantly higher and the age group 2–3 years was more affected. Admissions were significantly higher in 2021 than in 2020 and 2019, and the per-year hospitalization rate was lower in 2021 (84% vs. 93% in 2019), while the duration of admissions was similar. No difference was found in severity between 2019–2020–2021. In conclusion, after the COVID-19 pandemic, an increase in RSV cases in 2021 exceeding the median seasonal peak was detected, with the involvement of older children, while no difference was found in severity.

## 1. Introduction

The RSV (Respiratory Syncytial Virus) represents the most frequent viral cause of acute lower respiratory tract infection (LRTI) in children, with a worldwide distribution [1]. The RSV is an enveloped, spherical, negative-sense single-stranded RNA virus of the genus *Orthopneumovirus*, family *Pneumoviridae*. It is characterized by three membrane proteins called small hydrophobic (SH), attachment glycoprotein (G) and fusion protein (F). Its name comes from the large cells, known as syncytia, which form when infected cells fuse. The RSV is divided into A and B subtypes based on the protein G sequence. Subtype A appears to be related to more severe infections. Both subtypes circulate simultaneously during the annual epidemic season, although generally one of the two predominates each year [2]. The RNA-dependent replication cycle of this virus is significant because it is error prone and allows for the rapid generation of mutations, resulting in changes in the virulence of RSV and avoidance of potential future antiviral agents or vaccines [3].

The severe manifestations of RSV disease include pneumonia and bronchiolitis; the latter is usually self-limiting but, in infancy, it accounts for a significant number of hospitalizations and pediatric intensive care unit (PICU) admissions even in previously healthy, full-term children [1,4]. Bronchopulmonary dysplasia or other chronic respiratory conditions represent the main risk factors for severe infection, followed by age younger than 12 weeks, history of prematurity, male sex, crowding, formula feeding, congenital heart disease and immunodeficiency [5]. 

In 2015, the estimated global impact of RSV infections in children aged less than 5 years old was approximately 33 million LRTI episodes (uncertainty range: 21.6–50.3 million), 3.2 million hospitalizations (uncertainty range: 2.7–3.8 million) and 120,000 deaths (uncertainty range: 94,000–149,000) [6].

The RSV is a seasonal virus, characterized by variable epidemiology, depending on geographic area and climate. In the Northern hemisphere, the virus usually spreads between November and March with peak incidence in January/February, whereas in the Southern hemisphere, the RSV season is from June to September. 

The RSV epidemics are driven by a complex interaction between the climate, the host and the virus. Cold temperatures stabilize the RSV lipid envelope, humidity allows the deposition of heavy droplets on surfaces and, during cold and rainy periods, people gathering indoors facilitate RSV transmission. Moreover, it seems that new variants could drive RSV outbreaks, finding a susceptible population. The simultaneous isolation of different RSV strains in distant geographic areas supports the hypothesis that epidemics are driven by locally evolved variants [7,8].

The restrictive measures adopted worldwide against the SARS-CoV-2 virus (e.g., wearing masks, frequent hand washing and social distancing) produced a drastic reduction in respiratory pathogens in Europe, including RSV. In particular, bronchiolitis almost disappeared during the 2020/21 season, but a dramatic rebound was reported both in the Northern and Southern hemisphere during the 2021/22 season, largely due to RSV infection [9,10]. Moreover, surveillance data for 17 Countries showed delayed RSV epidemics in France (≥12 w) and Iceland (≥4 w) during the 2020/21 season [11]. 

Additionally, in Italy, we recorded a reduced circulation of RSV and other respiratory pathogens during the 2020/21 season detected by the Italian Influenza Surveillance Network (InfluNet) system [6]. 

Herein we present our epidemiological survey on RSV infection in children in Northwestern Italy in the last three years. 

The primary aim of our study was to evaluate RSV epidemiology during and after the COVID-19 pandemic peak in order to collect preliminary data on possible variations in the epidemiology and seasonality of RSV infection in our region. The secondary aim was to evaluate the clinical features of children with RSV infection.

## 2. Materials and Methods

Out of the 22 Pediatric Emergency Departments (EDs) of Piedmont, Northwestern Italy, that were invited to participate in this multicenter retrospective observational study, 15 accepted. Among the centers involved, 3 were major Pediatric EDs of hospitals provided with PICU and 12 were Pediatric EDs of hospitals without PICU.

All children <3 years old with RSV bronchiolitis and/or pneumonia admitted to the 15 Pediatric EDs between 1 September and 31 December 2021 were included and compared to those admitted in the same period of 2020 and 2019. The following diagnostic ICD codes (10th revision) were considered: 466.11, 079.6 and 480.1. Only children <3 years old at admission were enrolled, as this group has been reported to have the highest disease burden with RSV and in order to limit potential overlaps with other respiratory conditions. 

Epidemiological, clinical and microbiological data were retrospectively collected. The data were deidentified, recorded through a targeted registration form and transferred to a specifically designed database at the Coordinating Center before the analysis.

The following variables were included: demographic data (age and sex), clinical characteristics (gestational age if available, comorbidities, result for RSV search, coinfection by other microbiological agents including SARS-CoV-2 for the years 2020 and 2021, need for support with oxygen) and outcome (discharge home, admission to hospital, admission to PICU, transfer to another center); furthermore, both the overall and PICU length of stay (LOS) were recorded. 

Identification of the causal viral agent for bronchiolitis was carried out using nasopharyngeal aspirate or tracheal aspirate/bronchoalveolar lavage (in intubated patients) with an antigen test and/or with a multiplex polymerase chain reaction (PCR) assay (FilmArray^®^). SARS-CoV-2 determination was carried out using nasopharyngeal swab/aspirate by real-time reverse transcription-PCR (RT-PCR) in each patient with respiratory symptoms since the beginning of SARS-CoV-2 pandemic.

This study received ethical approval by the local Ethical Committee of the Coordinating Center (protocol number 0049894). 

The statistical analysis was performed using IBM SPSS Statistics 25.0 (IBM Corp. Armonk, NY, USA). Significance was set at *p* < 0.05. All p-values were 2 tailed. In the descriptive analysis, categorical variables are reported as absolute numbers and percentages, and continuous variables as mean, median and interquartile range (IQR), as appropriate. To compare the continuous variables of the study groups, a Student’s *t*-test was used. To evaluate the discrete variables, Pearson χ^2^ and correlation Fisher exact tests were performed, as appropriate. 

## 3. Results

During the three study periods (September–December 2019, 2020 and 2021), 1015 children <3 years old with RSV infection were enrolled: 100 in 2019, 3 in 2020 and 912 in 2021, respectively. The number of hospital admissions for each participating center is represented in Table 1.

Overall, in 2019, most RSV admissions in the four-month period under consideration occurred in December (93/100, 93%), while in 2021, most admissions occurred in November (480/912, 52.6%), as represented in Figure 1. The RSV peak was not accompanied by a high number of concomitant coinfections (0/93, 0/93 and 0/93, respectively, by SARS-CoV-2, Influenza virus A/B and bacteria in 2019 and 19/480, 0/480 and 16/480 in 2021). No conclusions can be drawn about the year 2020 given the low number of RSV-positive cases.

Considering only children tested at the major regional Pediatric Hospital and Coordinating Center of this study (Regina Margherita Children’s Hospital, Turin, Italy), laboratory results showed that in 2019, 20.0% of the children tested for RSV were positive, compared to 57.6% in 2021. Weekly test results for RSV, influenza and SARS-CoV-2 during the study periods of these two years are shown in Figure 2. 

Demographic and epidemiological characteristics of the total population are summarized in Table 2.

Males were more represented than females (54.3% vs. 45.7%), and the majority of children were ≤2 months old (36.9%) or 3–6 months old (37.3%). The median age was 2.8 months, IQR 1.5–6.3 months. Comparing 2019 to 2021, the median age was significantly higher in 2021 (4.1 months vs. 6.4 months, *p* = 0.018). Moreover, the age group between 2 and 3 years was more affected by RSV in 2021 than in 2019 (59/912, 6.4% vs. 2/100, 2.0%). 

Mean annual RSV admissions were significantly higher in 2021 than in 2020 (*p* = 0.003) and 2019 (*p* = 0.007); furthermore, there were significantly more admissions in 2019 than in 2020 (*p* = 0.0019).

Overall, 85.2% of patients needed to be hospitalized, and 6.3% required admission to the PICU. The risk factors for hospitalization are summarized in Table 3.

The per-year hospitalization rate (HR) was 93% in 2019 and 84% in 2021 (*p* = 0.0175); admissions in 2021 had a similar duration compared to admissions in 2019 (mean LOS: 5.77 days and 5.57 days, respectively). Hospitalization rates and LOS were different between the three major pediatric hospitals provided with a PICU and the twelve without a PICU (Table 4). The need for a PICU was similar in 2019 and 2021 (*p* = 0.67). 

The adjusted odds ratio (OR) of hospitalization and admission to PICU among the age groups is shown in Figure 3.

Moreover, the adjusted OR of hospitalization among age groups in 2021 compared to 2019 is shown in Figure 4.

By analyzing the risk factors for the need for oxygen supplementation, we found a significant relationship with prematurity (*p* = 0.00001) and age ≤ 2 months old (*p* = 0.00001). 

Regarding admissions to the PICU, we found that prematurity (*p* = 0.0001), age ≤2 months (*p* = 0.0001) and viral coinfections (*p* = 0.0002) were significantly related.

No difference was found in the severity of RSV cases (LOS—*p* = 0.59; admissions to PICU—*p* = 0.76; and mechanical ventilation—*p* = 0.33) between 2019-2020-2021. 

The number of patients who were transferred to one of the three hospitals with a PICU were, respectively, 3 in 2019 and 70 in 2021 (*p* = 0.10); no patient was transferred in 2020. On the contrary, from the main regional Children’s Hospital (Regina Margherita), 19 patients with RSV bronchiolitis (5.9% of the total cases admitted in 2021), of whom 17 were <6 months old, were moved to other hospitals due to a lack of beds between October and December 2021, compared to no transfers in the previous years. 

## 4. Discussions

In our study, we observed a significant increase in RSV-related hospital admissions in 2021 compared to previous years and a massive reduction in 2020 (under lockdown due to the pandemic) compared to 2019, as previously reported by other authors and an epidemiological report [10,12,13,14]. In 2020, the public health measures introduced during the COVID-19 pandemic had a great impact on the prevalence of respiratory virus infections other than COVID-19, such as influenza and RSV [9,15,16,17,18]. A subsequent increase in RSV cases exceeding the median seasonal peak of previous years was detected firstly in the Southern hemisphere [10,19], and then it was also confirmed in the Northern one [20]. For example, in New Zealand, the incidence rate during the bronchiolitis peak in 2021 was higher than the average of peaks in 2015–2019, but without more severe cases. 

Our data confirmed previous observations of different new epidemiological trends of RSV infections worldwide during the COVID-19 pandemic, not only in the Northern hemisphere [11,21,22,23,24,25,26] but also in the Southern one [25,27,28]. We observed an increase in RSV-related admissions in 2021, while no difference was found in the severity (LOS, admissions to PICU and rate of mechanical ventilation) of RSV cases between 2019, 2020 and 2021. The RSV disease is known to be a seasonal disease, responsible for large healthcare costs each year due to the high rates of hospitalization that it causes. Seasonality varies according to the geographical area, and RSV epidemics occur typically in autumn/winter in temperate regions: in the Northern hemisphere, the RSV season starts in November and ends in March-April, while in the Southern hemisphere, it lasts from May-June to September. In tropical countries, RSV circulates year round, with peaks during the rainy season [7,29,30]. After the COVID-19 pandemic, the circulation of other respiratory viruses, including RSV, seems to have at least partially changed [13]. We provided only a partial view of the seasonality of RSV infections over the past 3 years, because we took into consideration only the last four months of the three study years: nevertheless, there is evidence of a spike in cases in November in 2021, earlier than the usual peak between December and February in our latitudes. It is interesting to note that previous local studies with an observational study period extended from January to December did not observe any outbreaks of RSV bronchiolitis during spring and summer 2021 [31]. Another remarkable finding is the evidence of an important reduction in RSV cases after the advent of the Omicron variant in Italy, as previously reported by Halabi et al. also in the United States (USA) [32]. 

Better understanding of the seasonality of RSV and strict surveillance are required to improve the planning of health services. For example, redefining the start of the RSV season is important to ensure the optimal timing to deliver Palivizumab, a humanized monoclonal antibody (mAb) against RSV labeled for the prevention of severe respiratory infections due to RSV in high-risk children [33,34,35], as well as to schedule new possible strategies to prevent this infection. In the Piedmont region, considering what occurred in 2021, in the current year 2022, the start of Palivizumab prophylaxis occurred in October, with currently promising results.

During our atypical anticipated RSV peak in 2021, and despite the noticeable increase in RSV cases admitted to the EDs, we found a lower hospitalization rate compared to 2019, in contrast to what was previously described by Agha and Avner (two thirds of the cohort required admission) and Foley et al. (a 2.5-fold higher admission rate) during their atypical RSV surges. The hospitalization rate was lower especially in hospitals with a PICU compared to other hospitals: this finding confirms that the massive increase in the number of cases was not accompanied by any increase in the severity of the infection, compared to previous years.

Agha and Avner also reported that most of their hospitalized cohort required admission to the PICU. This is in contrast with our cohort, as only a minority of our patients required intensive care. Prematurity, age ≤2 months and viral coinfection was significantly related to PICU admission. Other studies in France [36], the USA [21,37] and Western Australia [19] have shown similar findings, with atypical surges of RSV not associated with more severe features, compared to previous seasons. A multicenter national survey with a wide geographical distribution in Italy reported the same results as well, with no difference in terms of disease severity according to proportions of intensive care admissions [20].

Interestingly, we noted an exceptionally low coinfection rate of SARS-CoV-2 unrelated to oxygen need or PICU admission. The same results have been described in previous studies, with no severe disease burden in the case of RSV and SARS-CoV-2 coinfection [7,19,22,38,39,40].

Anyway, the spike in cases in late 2021 has challenged the Italian National Health Care System: hospital reorganization, with the provision of temporary facilities to increase the number of inpatient beds, was necessary [20,31], particularly within referral centers equipped with a PICU. Despite this, in our setting in 2021, some patients with less severe diseases were transferred from the referral center to other hospitals due to ED overcrowding and a lack of beds, which rarely occurred in previous years. It is important to highlight this phenomenon, because in the future, we should be prepared to manage this kind of situation in case of a new RSV disease peak anticipated in time and unexpected in numbers. 

Newborns, younger infants and children >12 months old were found to be at a higher risk of hospitalization. In 2021, the risk of hospitalization for patients younger than 2 months old was particularly high (OR 5.8 vs. 3 in 2019). Another interesting finding is that the age group between 2 and 3 years was more affected by RSV infection in 2021 than in 2019, and for children older than 12 months of age, the odds of hospitalization was higher (OR 2.97 vs. 0.2 in 2019). These local data are consistent with what was reported by Saravanos and Foley [19,27,41] and Alrayes and Fourgeaud [36,38] in Australia, the USA and France. Such increased positivity rate may be secondary to immune-naive older children due to school closure and the suspension of all sports and recreational activities during the pandemic, as well as to increased testing rates in this population compared with previous years. The so-called immunity debt secondary to long periods of low exposure to RSV [10,19] has resulted in a higher than usual number of children >1 year old presenting to the Emergency Department for a picture of bronchiolitis/bronchitis/pneumonia and requiring hospitalization. The increased number of children with respiratory symptoms could have also resulted in more aggressive diagnostic testing for alternative infections, including RSV, to rule out COVID-19, as part of the measures to prevent the spread of COVID-19 infection. Moreover, in the case of hospitalizations, we were prone to testing older children for RSV more in order to group patients according to the identified respiratory agent and to avoid intrahospital transmission.

There is an urgent need to develop a robust management approach to detect, contain, deal with and potentially prevent new RSV outbreaks. They are not so easy to predict, considering the impact of extreme situations such as the recent pandemic and, last but not least, the evolving climate change. A great effort should be made in this direction, because timely and accurate data collection and national surveillance about RSV infection is essential to inform public health responses and to be ready to manage new outbreaks. 

Our study has some strengths. First of all, our observation provides information from different settings (hospitals with and without a PICU) in the same region: this could be useful to prevent the misuse of healthcare resources in case of future outbreaks, independently from national healthcare system assets. Moreover, we focused not only on newborns and infants with bronchiolitis, which have already been largely investigated, but also on older children with other respiratory features related to RSV infection. 

Our study also has some limitations. It is a multicenter study with a significant sample, but not all the centers invited to participate were included; moreover, only children who were admitted to the hospital were included, while outpatients treated by general practitioners were not considered. As a consequence, our results might not reflect the overall RSV positivity rate in our region. Moreover, the retrospective nature of the study could have limited available information. Another limitation is that RSV was tested with a PCR assay in some cases and with antigen testing in others depending on the different participating centers. Finally, we did not consider the entire epidemic period of RSV infection in Italy, which is extended until March-April, and our observation was limited to the last four months of each year.

## 5. Conclusions

The ongoing surveillance of RSV infection is necessary to recognize seasonal changes and to be ready to manage atypical spikes. After the COVID-19 pandemic, an increase in RSV cases in 2021 exceeding the median seasonal peak of previous years was detected. The unusual resurgence of RSV infection has also been reported in several other countries, as confirmed by our findings. Improved awareness of the timing of RSV epidemics ensures that optimal levels of protection are achieved in a cost-effective manner. In addition, better knowledge on the timing of RSV activity will allow healthcare providers to reorganize hospitals and manage resources to be ready to receive large numbers of patients with respiratory diseases outside the usual seasonal peaks, as was the case in 2021. Further studies that also consider the possible role played by climate change are needed to redefine the seasonality of RSV infection following the COVID-19 pandemic.

## Figures and Tables

**Figure 1 viruses-15-00280-f001:**
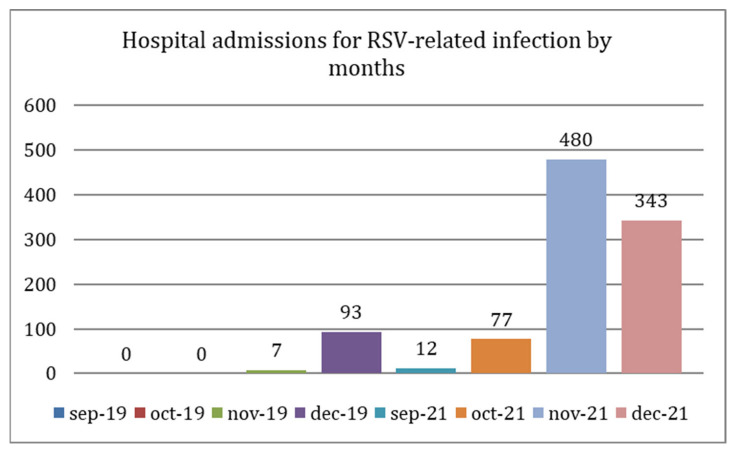
Hospital admissions for RSV-related infection by months (September-December) in 2019 and 2021: in 2019, the peak was recorded in December, while in 2021, the peak was recorded in November.

**Figure 2 viruses-15-00280-f002:**
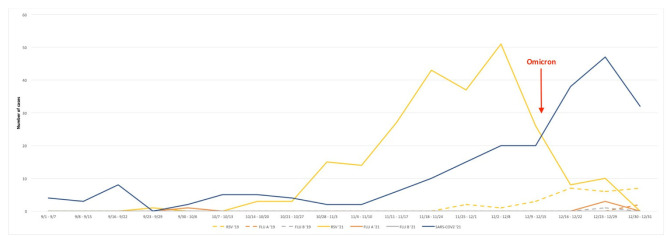
Weekly test results for RSV, influenza and SARS-CoV-2 during the study periods of 2019 and 2021 at the major regional Pediatric Hospital (Regina Margherita Children’s Hospital, Turin). Data from 2020 were excluded considering the extremely low number of positive cases. RSV: Respiratory Syncytial Virus; Flu: influenza virus; SARS-CoV-2: Severe Acute Respiratory Syndrome Coronavirus 2.

**Figure 3 viruses-15-00280-f003:**
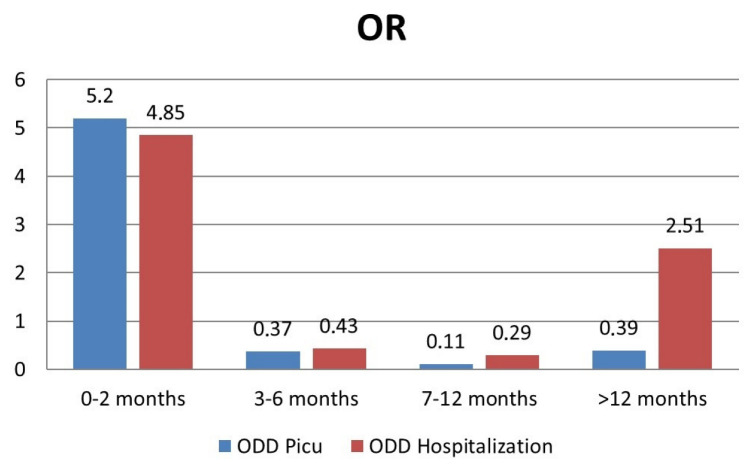
Risk of hospitalization and admission to PICU by age group.

**Figure 4 viruses-15-00280-f004:**
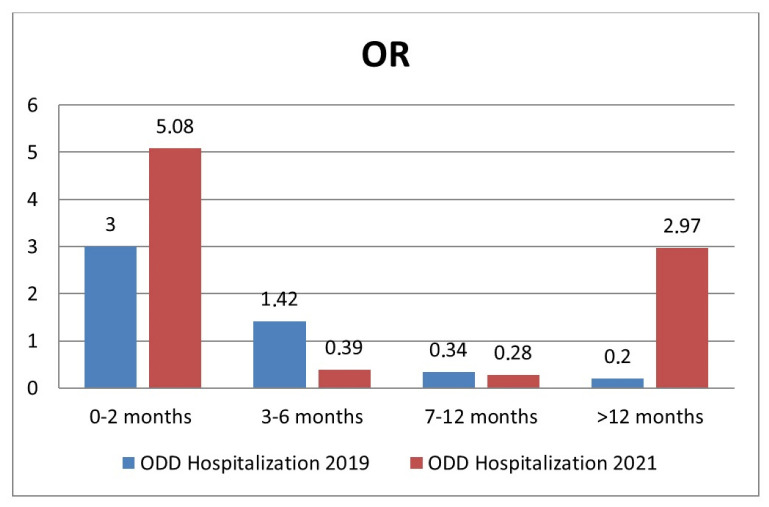
Risk of hospitalization by age group in 2021 compared to 2019.

**Table 1 viruses-15-00280-t001:** Hospital admissions (total number and percentage) for each participating center during the study period.

Hospital	Total Number of Cases	Frequency	2019	Frequency	2020	Frequency	2021	Frequency
Turin—Regina Margherita Children’s Hospital *	324	31.92%	27	27.00%	0	0.00%	297	32.57%
Alessandria *	132	13.00%	9	9.00%	2	66.67%	121	13.27%
Asti	97	9.56%	10	10.00%	0	0.00%	87	9.54%
Rivoli	84	8.28%	15	15.00%	0	0.00%	69	7.56%
Novara *	81	7.98%	3	3.00%	0	0.00%	78	8.55%
Cuneo	50	4.93%	2	2.00%	0	0.00%	48	5.26%
Turin—Martini	49	4.83%	13	13.00%	0	0.00%	36	3.95%
Ciriè	41	4.04%	3	3.00%	0	0.00%	38	4.17%
Turin—Maria Vittoria	36	3.55%	7	7.00%	0	0.00%	29	3.18%
Pinerolo	30	2.96%	3	3.00%	0	0.00%	27	2.96%
Chivasso	26	2.56%	0	0.00%	0	0.00%	26	2.85%
Borgomanero	20	1.97%	0	0.00%	1	33.33%	19	2.08%
Verbania	18	1.77%	1	1.00%	0	0.00%	17	1.86%
Chieri	15	1.48%	3	3.00%	0	0.00%	12	1.32%
Biella	12	1.18%	4	4.00%	0	0.00%	8	0.88%
Total	1015	100.00%	100	100.00%	3	100.00%	912	100.00%

* Hospitals equipped with pediatric intensive care units (PICUs).

**Table 2 viruses-15-00280-t002:** Demographic and epidemiological characteristics of the cohort.

		Number (%)	2019	2020	2021
Sex	Male	551 (54.3%)	54 (54%)	2 (66.6%)	495 (54.3%)
	Female	464 (45.7%)	46 (46%)	1 (33.3%)	417 (45.7%)
Age	≤2 months	375 (36.9%)	32 (32%)	0	343 (37.6%)
	3–6 months	379 (37.3%)	51 (51%)	1 (33.3%)	327 (35.9%)
	7–12 months	119 (11.7%)	13 (13%)	1 (33.3%)	105 (11.5%)
	13–24 months	81 (7.9%)	2 (2%)	1 (33.3%)	78 (8.5%)
	2–3 years	61 (6.4%)	2 (2%)	0	59 (6.4%)
Coinfections	SARS-CoV-2	18 (1.8%)	-	0	18 (2%)
	Other coinfection	57 (5.6%)	4 (4%)	1 (33.3%)	52 (5.7%)
Risk factors	Prematurity	119 (11.8%)	5 (5%)	0	114 (12.5%)
	Immune deficiency	2 (0.2%)	0	0	2 (0.2%)
	Heart disease	33 (3.3%)	0	0	33 (3.6%)

**Table 3 viruses-15-00280-t003:** Risk factors for hospitalization.

		Admissions (Number)	*p*	Children Admitted to PICU (Number)	*p*	Children Who Needed Oxygen Supplementation (Number)	*p*
Sex	Male Female	469 396	0.93	36 28	0.79	379 307	0.38
Age	≤2 months	356	0.0001	47	0.0001	301	0.0001
	3–6 months	297		12		223	
	7–12 months	80		1		60	
	13–36 months	132	0.0047	4		102	
Coinfections	SARS-CoV-2	17	0.49	1	1	15	0.2
	Other coinfection	56	0.0004	10	0.0002	37	0.66
Risk factors	Prematurity	111	0.006	19	0.0001	97	0.0001
	Immune deficiency	28	1	4	0.15	22	1
	Lung disease	11	1	0	1	10	0.56
	Heart disease	2	1	0	1	2	1

**Table 4 viruses-15-00280-t004:** Comparison in LOS and HR between hospitals with PICU and hospitals without PICU in 2019 and 2021.

	Hospitals with PICU	Hospitals without PICU	*p*
Mean general LOS * mean LOS in 2019 mean LOS in 2021	6.74 days	4.68 days	<0.00001
	6.48 days	4.48 days	<0.00001
	6.75 days	4.72 days	<0.00001
Mean general HR ** mean HR in 2019 mean HR in 2021	0.79	0.91	<0.0001
	0.91	0.94	0.54
	0.78	0.91	<0.0001

* LOS: length of stay; ** HR: hospitalization rate.

## Data Availability

The collected data are available from the corresponding author upon reasonable request.

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
