# Peer review of "The Importance of RSV Epidemiological Surveillance: A Multicenter Observational Study of RSV Infection during the COVID-19 Pandemic"

_viruses, 2023, doi:10.3390/v15020280_

Round 1
Reviewer 1 Report
Not much to add to this - a simple observation study that is well-described with limitations s highlighted. Many labs elsewhere are also seeing this viral rebound phenomena - not just with RSV but other resp viruses also.
Author Response
Thank you for your review.
Reviewer 2 Report
The manuscript by Pruccoli et al presented an observational study of RSV infection during the COVID-19 pandemic from multiple centers. This study has public health relevance as RSV’s huge rebound recently and high prevalence among infants and children. These areas regarding the manuscript that can be improved are listed below.
1. The introduction section should introduce the viral structure of RSV in the background.
2. In table 1, the commas should be replaced by decimals in the table.
3. Lines 149-151: are the admissions only caused by RSV or could also be co-infected with COVID-19 and/or influenza A/B and/or other bacteria?
4. Figure 2: Please show the percentage and numbers on the bars in figure 2.
5. Line 156: is “Regina Margherita Children’s Hospital” one of the 15 EDs? If not, how the data was collected, and what’s the relevance to the RSV admission at the 15 EDs?
6. Figure 3: Please add the label for the y-axis.
7. Lines 177-179: age distribution from 2019, 2020, and 2021 were not presented in table 2. Please include the demographic characteristics of cohorts from different years in table 2 as well.
8. Line 179: there are only 912 children enrolled in 2021, how would the RSV rate for ages 2-3 be calculated using 915 for 2021?
9. Table 3: Female data was not missing in the table, and the numbers don't add up.
10. Figure 4: please replace the commas with decimals.
Author Response
Thank you for your review.
As requested:
- We added in the introduction section a paragraph about the viral structure of RSV (Lines 66-76) with two additional references (number 2 and 3);
- We modified Table 1, replacing commas with decimals
- Lines 149-151 (now 159-161): we added the presence of co-infections at the time of peak hospitalizations for RSV (in December in 2019 and November in 2021)
- Figure 2 (now Figure 1): we showed the numbers on the bars
- Line 156-157 (now 166-167): We specified in Table 1 that “Regina Margherita Children’s Hospital” is one of the 15 EDs (it is the main pediatric hospital in Turin and the coordinating Center of this study) and also in the main text
- Figure 3 (now Figure 2): we added the label for the y-axis
- Lines 177-179 (now 185-187): we included in Table 2 age distribution from 2019, 2020, and 2021 and the demographic characteristics of cohorts from different years as well
- Line 179 (now 187): there is a typing error, the correct denominator for the year 2021 is 912 and not 915, we thank you for the correction
- Table 3: we added female data about hospitalization compared to male data: admissions, children admitted to PICU, children who needed oxygen supplementation (data are different from those of Table 2, regarding all the cohort including patients discharged from the EDs and not hospitalized).
- Figure 4 (now Figure 3): we replaced the commas with decimals
Reviewer 3 Report
Reviewer comments:
In general, the paper is well thought out and written. There are a few minor points below that should be taken into account. The major criticism is that authors lump essentially all of the analyses together for the duration of the evaluation period. The only place they compare the years is in Table 4. It could be very instructive to know how (or if) the disease characteristics changed between 2019 and 2021 given the circulation of SARS-Cov-2 in 2021.
Materials and Methods
1. Why use 3 years as the upper limit rather than 2 years.
2. Why select 9/1 to 12/31 as study window if the virus peaks in Jan/Feb?
3. How was this testing standardized across the multiple centers, particularly since this was retrospective?
Results
1. Table 1:
a. If you indicate which hospitals are PICU vs. non-PICU you can easily delete Table 1 since both graphics contain the same information.
b. There needs to be additional columns showing the breakdown by year. You can then include the relevant p-values for the appropriate comparisons.
2. Figure 2:
a. If the breakdown by year is included in Table 1, figure 2 is not needed.
3. Line 156:
a. I assume Regina Margherita Children’s Hospital is in Turin. If so, it should be indicated.
4. Lines 157 and 158:
a. Is the 20% and 57.6% of the patients admitted for RSV and included in the study population or is it from a different population. Please clarify.
5. Figure 3:
a. Need to correctly label the y-axis
6. Table 2:
a. Need to show the demographic characteristics of by year. This is particularly important since the years in question are before and after the pandemic and may be quite different.
7. Table 3:
a. Need to show a breakdown by year. Again, this is important since the years in question are before and after the pandemic and may be quite different.
8. Figure 4:
a. An overall OR calculation is fine but it should also be evaluated by year.
Author Response
Thank you for your review.
As requested:
Materials and Methods:
- We used 3 years as the upper limit rather than 2 years to focus our study not only on newborns and infants, which have already been largely investigated, but also on older children with other respiratory features related to RSV infection.
- We select 9/1 to 12/31 as study window because of the significant RSV-peak recorded in November 2021, which was not detected in early 2021 neither at the beginning of the following year, 2022. We therefore decided to focus only on the last 4 months of the year, comparing them with the same period in 2019 (pre-COVID) and 2020 (under lockdown). In the future, it might be interesting to investigate the entire epidemic season of the virus in Italy, if not even the entire current year in comparison with previous years, to study possible changes in seasonality of RSV infection after COVID-19 pandemic.
- This is a limitation of our study as specified in the text; the RSV was tested in some cases by PCR assay, in some others by antigen testing, depending on the different participating centers.
Results:
- We deleted Figure 1 and modified Table 1, replacing commas with decimals, indicating which hospitals are PICU vs. non-PICU, specifying that Turin - Children’s Hospital is “Regina Margherita” and adding additional columns showing the breakdown by year.
- Figure 2 (now Figure 1): we would prefer not to eliminate this figure if you agree with it
- Line 156-157 (now 166-167): We specified in Table 1 that “Regina Margherita Children’s Hospital” is one of the 15 EDs (it is the main pediatric hospital in Turin and the coordinating Center of this study) and also in the main text
- Lines 157 and 158 (now 166-169): these percentages (20% in 2019 and 57.6% in 2021) refer to children included in the study population, specifically from the subgroup tested at the Coordinating Center of the study (Regina Margherita Children’s Hospital). To clarify this concept, we changed the sentence as follows: “Considering only children tested at the major regional Pediatric Hospital and Coordinating Center of this study (Regina Margherita Children’s Hospital), laboratory results show that in 2019, 20% of children tested for RSV were positive, while in 2021, 57.6%.”
- Figure 3 (now Figure 2): we added the label for the y-axis
- Table 2: we included age distribution from 2019, 2020, and 2021 and the demographic characteristics of cohorts from different years as well
- Table 3: we decided not to show a detailed breakdown by year in Table 3, considering the extremely low number of patients in 2020, the wide difference in numbers between 2019 and 2021 that makes it difficult to compare results, and the absence of statistically significant data about severity of infection (LOS - p 0.59 - admissions to PICU - p 0.76 - and mechanical ventilation - p 0.33) between 2019-2020-2021, as reported in the main text (Lines 209-210)
- Figure 4 (now Figure 3 and Figure 4): we replaced the commas with decimals and we have considered not only the overall OR calculation, but also by year comparing 2021 to 2019.